# The Role of Thickness Control and Interface Modification in Assembling Efficient Planar Perovskite Solar Cells

**DOI:** 10.3390/molecules24193466

**Published:** 2019-09-24

**Authors:** Weifu Sun, Kwang-Leong Choy, Mingqing Wang

**Affiliations:** 1Institute for Materials Discovery, University College London, London WC1E 7JE, UK; weifu.sun@bit.edu.cn (W.S.); MINGQING.WANG@UCL.AC.UK (M.W.); 2State Key Laboratory of Explosion Science and Technology, Beijing Institute of Technology, Beijing 100081, China

**Keywords:** perovskite solar cell, titanium dioxide, surface treatment, titanium tetrachloride

## Abstract

Perovskite solar cells (PSCs) have achieved tremendous success within just a decade. This success is critically dependent upon compositional engineering, morphology control of perovskite layer, or contingent upon high-temperature annealed mesoporous TiO_2_, but quantitative analysis of the role of facile TiCl_4_ treatment and thickness control of the compact TiO_2_ layer has not been satisfactorily undertaken. Herein, we report the facile thickness control and post-treatment of the electron transport TiO_2_ layer to produce highly efficient planar PSCs. TiCl_4_ treatment of TiO_2_ layer could remove the surface trap and decrease the charge recombination in the prepared solar cells. Introduction of ethanol into the TiCl_4_ aqueous solution led to further improved open-circuit voltage and short-circuit current density of the related devices, thus giving rise to enhanced power conversion efficiency (PCE). After the optimal TiCl_4_ treatment, PCE of 16.42% was achieved for PSCs with TiCl_4_ aqueous solution-treated TiO_2_ and 19.24% for PSCs with TiCl_4_ aqueous/ethanol solution-treated TiO_2_, respectively. This work sheds light on the promising potential of simple planar PSCs without complicated compositional engineering and avoiding the deposition and optimization of the mesoporous scaffold layer.

## 1. Introduction

Perovskite materials have attracted attention due to the unprecedented success of perovskite solar cells (PSCs) [1,2,3], as well as other optoelectronic applications, including light-emitting diodes (LEDs) [4], laser diodes [5] and photodetectors [6], etc. The record certified power conversion efficiency (PCE) has surpassed 24.2% [7] in less than one decade since their first application in dye-sensitized solar cells [8,9,10,11,12,13,14,15,16] from initial 3.8% in 2009. A variety of approaches have been developed to enhance the PCE, such as interfacial engineering [10], composition engineering [11], and morphology control through solvent engineering [9]. The solar-to-electrical energy conversion efficiency of PSCs has been enhanced but is critically contingent upon the high-temperature annealed mesoporous titania (TiO_2_). Titanium tetrachloride (TiCl_4_) treatment of the compact TiO_2_ layer and subsequently, annealing/sintering also play an important role in enhancing the PCEs [17,18,19,20]. The annealing process converts species from the TiCl_4_ solution to TiO_2_ crystallites on the surface of the TiO_2_ nanocrystalline films. The TiO_2_ crystallites derived from the TiCl_4_ treatment are reported to improve the cell performance owing to an increased surface area of the films, which increases perovskite loading and, correspondingly, light-harvesting efficiency.

It has, however, been argued that the extent to which the device performance improves is complicated by the effect that TiCl_4_ treatment has on the TiO_2_ conduction band edge position, charge injection, transport, and recombination [21,22,23]. It has been discovered that the surface traps, which are predominately located on the TiO_2_ surface [22,24], are capable of limiting transport and recombination. In addition, the resulting TiO_2_ layer formed on the surface of the TiO_2_ nanocrystalline film is expected to influence the density and distribution of surface traps. Thus, the electron transport and recombination kinetics can be influenced [25,26]. It is supposed that the recombination rate will increase with an increase in the number of surface traps and high surface roughness of a film [22]. However, the controversial conclusion has also been drawn that the TiCl_4_ treatment decreases the recombination rate and improves the charge injection efficiency but has a trivial impact on the light-harvesting and electron transport [18,23]. Thus far, the effects of varying the TiCl_4_-treatment conditions in terms of the concentration on the TiO_2_ film morphology, light-harvesting, and performance of dye-sensitized solar cells (DSSCs) have been explored [27,28,29]. It has been found that an intermediate TiCl_4_ concentration (e.g., 40 mM) gives rise to the best device performance, which compromises various competing factors, such as electron transport, recombination, and light-harvesting [27]. TiCl_4_ treatment has been widely used in perovskite solar cells [30,31], typically with a fixed concentration of 40 mM aqueous solution at 70 °C for 30 min. In this study, the addition of ethanol to this solution is employed to better control the hydrolysis rate of TiCl_4_ and the particle growth of the TiO_2_ formed from the hydrolysis.

In this work, a simple device architecture-planar perovskite solar cell (PSC) based on the methylammonium lead iodide (MAPbI_3_) perovskite acting as a light harvester and TiO_2_ acting as an electron transport layer (ETL) is employed. Perovskite was synthesized using a one-step approach by mixing MAI and PbI2 precursors in solvents and was characterized using scanning electron microscopy (SEM), X-ray diffraction (XRD), ultraviolet-visible (UV-vis) and photoluminescence (PL) spectra. The thickness was optimized by controlling spin-coating speed and monitoring the photovoltaic performance of the assembled PSCs. Finally, by keeping the concentration of TiCl_4_ and reaction temperature constant and varying the reaction time, the effect of the TiCl_4_ (water/ethanol) treatment time on the surface morphology, light-harvesting, and the device performance of the assembled PSCs was investigated. 

Note that as the effects of TiCl_4_ treatment on charge transport kinetics have been studied elsewhere in dye-sensitized solar cells (DSSCs) systems [21,22,23], little attention has been paid to it in the present work. Further, though TiCl_4_ treatment and thickness of 100–150 nm were frequently documented, the quantitative analysis of their role or the origin has been largely ignored. To this end, in the present study, efficient PSCs were successfully fabricated and the roles of facile thickness control and surface modification quantitatively characterized.

## 2. Results and Discussion

### 2.1. Characterisation of Perovskite Film 

The morphology and phase structure of the synthesized methylammonium lead iodide (MAPbI_3_) were characterized using SEM and XRD. As observed from Figure 1a,b, a uniform, pinhole-free, and compact film of perovskite has been successfully achieved with grain size 200–400 nm. The MAPbI_3_ perovskite and compact TiO_2_ films on fluorine-doped tin oxide (FTO) were identified using XRD patterns, as displayed in Figure 1c. A set of preferred orientations located at 14.10°, 28.45°, 31.85°, 40.60°, and 43.15° can be ascribed to the (110), (220), (310), (224), and (330) planes of the MAPbI_3_ perovskite tetragonal structure, respectively. In addition, other minor peaks present at 2θ = 20.00°, 23.50°, 24.50°, 34.95°, 50.25°, and 52.55°are characteristics of the (200), (211), (202), (312), (404), and (226) planes, respectively [32]. The appearance of a series of new diffraction peaks that are in good agreement with literature data on the tetragonal phase of the CH_3_NH_3_PbI_3_ perovskite [33]. These results indicate that all perovskite films are of high phase purity. The spin-coated electron transport layer (ETL) TiO_2_ deposited in both glove boxor in ambient environment exhibits a minor peak at 2θ = 25.25°, indicating the formation of anatase TiO_2_. The measured optical properties including UV-vis absorption and photoluminescence (PL) spectra of the synthesized CH_3_NH_3_PbI_3_ perovskite are shown in Figure 1d. Band-edge absorption and emission are observed at 780 and 787 nm, respectively, corresponding to E_g_ of 1.590 and 1.576 eV (*E*_g_=1240/λ_band-edge_). The full width at half maximum (FWHM) of the PL spectrum is approx. 52 nm which is in agreement with published results elsewhere [34].

### 2.2. Thickness Influence of Electron Transport Layer TiO_2_ on Device Performance

The thickness of the electron transport layer (ETL) TiO_2_ was controlled by varying the spin-coating speed, followed by the TiCl_4_ in mixed deionized water/ethanol solution at 70 °C for 30 min. The thickness of the spin-coated ETL was measured using an optical profilometer and the J-V curves of the prepared devices were recorded. The average photovoltaic parameters of the device performance based on different thickness of ETL are summarized in Table 1. As observed from Figure 2a,c, both the open-circuit voltage (*V*_oc_) and fill factor (FF) did not change significantly with a thickness of TiO_2_. In contrast, the short-circuit current density (*J*_sc_) increased from a value of 19.42 ± 3.25 mA cm^−2^ to 20.21 ± 1.54 mA cm^−2^ between a thicknesses of 56.1 nm and 152.2 nm before decreasing to 19.66 ± 1.48 mA cm^−2^ at a thickness of 244.0 nm (Figure 2b). The statistical distributions of the photovoltaic parameters of the assembled 15~20 PSCs are displayed in Appendix A. The champion devices in each case had PCE of 17.51%, 18.32%, and 17.72%, respectively. The J-V curves are shown in Figure 3a.

Electrical impedance spectroscopy (EIS) was used to characterize the assembled PSCs. Nyquist plots were recorded in the frequency range from 1.5 MHz to1 kHz at open-circuit voltage. The equivalent circuit used is shown in the inset of Figure 3, where the R_s_ value corresponds to the resistance of the conducting substrate, electrode, and wires, while R_ct_ is related with the charge transport resistance of the interface and in the hole transport layer and electron transport layer. Under the above EIS test condition, the high-frequency arc is attributed to the charge transport resistance R_ct_. The differences in the high-frequency arc are mainly due to the charge transport resistance of TiO_2_ electron transport layer. As shown in Figure 3b, the device with TiO_2_ thickness of 152 nm has the smallest charge transfer resistance, as indicated by the smallest arc in the Nyquist plot. This can explain why the 152 nm ETL layer PSC has the largest short-circuit current density. 

The relationship between the thickness and charge transfer can be explained in terms of the electrical conductivity of inorganic semiconductor-blocking layer TiO_2_. As TiO_2_ is the electrically conducting layer and hole blocking layer, too thin or too thick TiO_2_ layer will inhibit the electrical conductivity. If the thickness of TiO_2_ is too small (e.g., 30 nm), discontinuities lead to current leakage and device failure [35], too thick and the charge transfer resistance will increase sharply and result in lower current density. Based on this rationale, although only three thicknesses have been explored, it can be expected that further increases in thickness (above 250 nm) will not only decrease the short-circuit density, but also lead to the total thickness probably surpassing the diffusion length of MAPbI_3_ [36]. Since the electron/hole diffusion length is an important parameter in solar cells which is directly related to the diffusion coefficient *D* and the photoluminescence decay rate *τ*_e_ in the absence of any quencher material [34,37]. The high device efficiency critically depends on efficient exciton (electron-hole pairs) generation and dissociation, and charge carrier transfer and recombination [38]. If the absorber layer is too thick, the generated electron and holes may experience charge carrier recombination before being collected. 

In order to further clarify this, photoluminescence quenching tests for c-TiO_2_ thickness-dependent films (glass/c-TiO_2_/perovskite) as compared to pure perovskite (glass/perovskite) have also been considered. As compared to pure perovskite, the TiO_2_/perovskite layer shows enhanced PL quenching effect, as observed in Figure 3c. With the increase of the thickness of the TiO_2_ layer, the quenching effect of PL spectra also increases. However, when the thickness increases to more than 152 nm, the difference in the PL quenching effect is not so obvious. The dependence of resistivity of TiO_2_ film on thickness can be controversial, which are affected and complicated by factors such as the microstructure (e.g., grain size, void size), processing pressure, etc. It is discovered that the resistivity of micro-porous magnetron-sputtered thin TiO_2_ films decreases with increasing film thickness [39], whereas the resistivity of nanometer TiO_2_ thin film enhances with the increase of the thickness of the films [40], varying from conductor, semiconductor to non-conductor. This opposite difference in trend is associated with the different microstructure, either porous or densely compact structure. In our case, the ETL consists of the bottom compact TiO_2_ layer and the top relatively micro-porous layer, which compounds the difficulties in achieving the monotonic trend. Besides, the Jsc observed from the single three J-V curves in Figure 3a indeed displays a monotonic decrease with increasing thickness. However, these single three curves do not have statistical meaning. As observed from Table 1, the average Jsc of PSC devices based on 152 nm thickness gives a higher value of 20.21 ± 1.54 than other devices based on the thickness of 56.10 and 244.0 nm. The rationale of displaying these three curves in Figure 3a is because each curve is the champion device based on the three different thicknesses of ETL. 

### 2.3. Influence of Treatment Time of Titanium Tetrachloride on Device Performance

The post-treatment of compact TiO_2_ using TiCl_4_ has been widely used in solar cells, including dye-sensitized solar cells (DSSCs) [41,42,43,44] and PSCs [42,44]. The effect of concentration of TiCl_4_ aqueous solution on the morphology of the formed porous TiO_2_ has been studied previously [43,45]. With an intermediate TiCl_4_ concentration (15−50 mM), the surface area of the TiO_2_ films increases, resulting in an increase of light-harvesting and overall power conversion efficiency. While at a high TiCl_4_ treatment concentration (500 mM), although light scattering in the film in the long-wavelength region of the visible spectrum was enhanced, the average pore size of the film became narrower, resulting in slower transport and loss of cell performance. Therefore, 40 mM of TiCl_4_ solution was chosen in this work. The underlying mechanism of TiCl_4_ treatment to enhance efficiency has been explored in terms of charge carriers’ dynamics [41,42,43,44]. TiCl_4_ treatment can engineer the band edge alignment by downward shifting the conduction band edge of TiO_2,_ which leads to improved electron injection into TiO_2_ and high photocurrent [41]. In addition, TiCl_4_ treatment can passivate the surface traps of TiO_2_ and decrease the electron/hole recombination rate [42]. Note that although some TiO_2_ nanoparticles have been produced on the surface of TiO_2_ compact layer after surface treatment, only a small amount has been introduced. 

As the TiCl_4_-treated compact TiO_2_ was not subsequently subjected to the spin coating of a commercial paste diluted in ethanol to deposit the mesoporous layer of either 20-nm or 30-nm sized particles as typically employed in assembling mesoporous PSCs [8,13], it can be classified as a planar-structured PSC. In this work, to slow the hydrolysis process and control the size of the formed TiO_2_ nanoparticles, a mixed solution of deionized water and ethanol was used as a reaction solvent medium of TiCl_4_ hydrolysis to pretreat compact TiO_2_ layer [45,46]. The effect of different treatment times ranging from 0, 30, 60, and 90 min on the morphology and the device performance of the finally assembled PSCs were also investigated using SEM, atomic force microscopy (AFM) and J-V curves.

As observed from SEM images in Figure 4a, one compact TiO_2_ film formed on the FTO glass, but some cracks were also present. During the course of the deposition of compact TiO_2_, the spin-coated TiO_2_ on FTO glass using 0.2 M TiAcAc as precursor solution was subsequently subjected to thermal annealing at 500 °C for 45 min. If the heating rate from circa 125 °C to 500 °C was not slow enough, the solvent 2-propanol evaporated rapidly, which could have probably lead to the crack formation. To remedy these cracks, TiCl_4_ solution (ethanol/water) was used to post-treat the compact TiO_2_ layer at 70 °C for 30, 60, and 90 min, as shown in Figure 4b–d, respectively. The surface morphology became gradually smoother with increasing treatment time. As observed from the corresponding AFM images, the surface roughness decreased from the initial RMS of 25.45 nm to 15.05, 12.87 and 11.72 nm, respectively. Apart from the passivation of surface state, TiCl_4_ treatment also has the following benefits: On one hand, it can remedy the cracks formed by forming porous TiO_2_ film on the FTO glass if there is a very thin compact TiO_2_ layer onto FTO glass. This will contribute to the avoidance of direct contact between FTO and perovskite and the corresponding current leakage. On the other hand, the formed cracks would result in the relatively higher surface roughness. But it should be noted that the loading of perovskite and interaction between TiO_2_ layer and perovskite are dependent on the surface roughness since the porous layer is prone to absorb perovskite readily and create a close contact between TiO_2_ layer and perovskite.

For the purpose of comparison, the conventional recipe of TiCl_4_ aqueous solution was also employed to perform TiCl_4_ treatment at 70 °C for 30 min at a concentration of 40 mM. The SEM and AFM images of the resulting TiO_2_ particles on the surface of nanocrystalline TiO_2_ are shown in Appendix A. It can be observed that agglomeration of TiO_2_ nanoparticles has occurred and the surface roughness has dropped from 25.45 nm for the untreated TiO_2_ surface to 20.07 nm, although this is still much higher than that of TiO_2_ surface treated using a water/ethanol solution (15.05 nm) as shown in Figure 4b. The effect of the different TiCl_4_ solutions on device performance was also explored and the main photovoltaic parameters are summarized in Table 2. In striking contrast, both *V*_oc_ and *J*_sc_ of the devices with pure water solution TiCl_4_ treatment have dropped from 1.05 ± 0.01 and 20.21 ± 1.54 mA cm^−2^ for devices fabricated with water/ethanol to 0.98 ± 0.01 V and 16.26 ± 2.54 mA cm^−2^, although the FF improved slightly. The ethanol in the recipe of TiCl_4_ solution can efficiently inhibit the agglomeration of TiO_2_ nanoparticles and control the surface roughness of TiO_2_ film.

As observed from Figure 5a,c, both the open-circuit voltage and fill factor of the solar cells increase with increasing TiCl_4_ treatment time from 0, 30, 60, to 90 min, with the *V*_oc_ changes from 1.03 ± 0.01 to 1.05 ± 0.01, 1.06 ± 0.02, 1.06 ± 0.02 V and the corresponding FF from 51.74 ± 3.71% to 62.67 ± 3.16%, 63.00 ± 5.00% and 67.92 ± 3.48%, respectively. In contrast, the short-circuit current density first increases from 19.39 ± 0.70 mA cm^−2^ based on bare TiO_2_ to a peak value of 20.21 ± 1.54 mA cm^−2^ for 30 min TiCl_4_ treatment, before decreasing to 19.82 ± 2.28, 15.73 ± 1.74 mA cm^−2^ for 60 and 90 min TiCl_4_ treatment, respectively. The statistical distributions of the photovoltaic parameters of the assembled 15~20 PSCs based on different treatment times of TiCl_4_ are displayed in Appendix A. 

Low and high magnification SEM images of the assembled PSCs cross-sections are shown in Figure 6a,b. The thickness of the perovskite film and hole transport layer (HTL) are estimated to be 350 and 400 nm, respectively. The champion devices in each case give the best power conversion efficiency (PCE) of 12.75%, 19.24%, 17.24%, and 15.36%, respectively, and the J-V curves are shown in Figure 6c. The PCE of 19.24% is competitive with the best planar perovskite solar cells reported to date (17–21%) [4,47,48,49]. Most of these works either use mixed cation and halide using the concept of composition engineering or exploit precise stoichiometry to enhance the device performance, herein facile approach has been adopted to achieve high efficiency by simply optimizing TiCl_4_ treatment. An optical image of the synthesized perovskite film on TiO_2_/FTO glass is displayed in Figure 6d.

Electrical impedance spectroscopy was used to characterize the assembled PSCs. Note that, under other identical conditions, the surface treatment using TiCl_4_ water/ethanol will produce loose-structured TiO_2_ or porous (hydrolysis) atop the blocking-layer TiO_2_, while the electrical conductivity of porous TiO_2_ is even worse than the compact blocking layer TiO_2_, even if the porous TiO_2_ has been annealed at high temperature. The creation of porous TiO_2_ aims to readily arrest the perovskite and improve the interface between TiO_2_ and perovskite layer. However, even if a small amount of porous TiO_2_ has been introduced during surface treatment, the type of PSCs still falls within planar PSCs because the mesoporous TiO_2_ with obvious thickness of more than 100 nm has not been spin-coated in present work.

As shown in Figure 7a, the device based on TiCl_4_ treatment time of 30 min gives the smallest charge transfer resistance, as indicated by the semi-circle in Nyquist plot. This can explain why the PSC with the intermediate treatment time of 30 min gives rise to the largest short-circuit current density. Another factor is that the longer the treatment time, the higher the thickness of the porous TiO_2_ layer. Thus, this might lead to the lower transmittance of the substrate and deteriorate the light-harvesting by the perovskite film atop the TiO_2_/FTO substrate. As observed from Figure 7b, at this step, after being subjected to surface treatment, the transmittance of FTO glass /TiO_2_ sample at the wavelength of 600 nm decreased from 71.9% to around 65%, 63.5%, and 63.2% after 30, 60, and 90 min treatment, respectively. This value further decreases to 62.1% when performing TiCl_4_ treatment using an aqueous solution. Our attention here was paid to the trend of transmittance with increasing the treating time, but not to the magnitude of transmittance. It should be noted that the FTO glass/ TiO_2_ will be further annealed at high temperature and so the transmittance will not keep unchanged and probably even improve because of the higher degree of crystallinity. 

## 3. Experimental 

### 3.1. Chemicals

Anhydrous ethanol, 2-propoanol (isopropanol), acetone, acetonitrile, chlorobenzene, *N*, *N*-dimethylformamide, titanium diisopropoxidebis(acetylacetonate) (75% solution in 2-propanol), zinc powder, hydrochloric acid (37wt%) and Spiro-MeOTAD (99%) were all purchased from Sigma-Aldrich and used as received. Methylammonium iodide was from Dyesol. Lead (II) iodide was purchased from Alfa Aesar. FTO glass was purchased from Sigma-Aldrich (7 Ω/γ, TEC 7, Shanghai, China). Silver pellet (99.99%) was purchased from Kurt J Lesker Company Ltd (Deutsch, German).

### 3.2. Device Fabrication

FTO glass was etched using zinc powder and hydrochloric acid (25 wt. % HCl), before being sequentially washed with deionized water, acetone, and isopropanol under ultra-sonication for 15 min each, respectively, then subjected to air plasma for 10–15 min. Following this, a 0.2 M titanium diisopropoxidebis (acetylacetonate) (TiAcAc) solution was spin-coated on the pre-patterned FTO glass to achieve a compact layer of TiO_2_ with desired thickness ranging from 50 to 250 nm. Subsequently, the FTO glass was immediately heated to 125 °C for 10 min, then heated to 500 °C for 45 min before being cooled to room temperature. The resultant compact TiO_2_ was immersed into either 40 mM TiCl_4_ ethanol/water or aqueous solution at 70 °C for a range of times before being sintered at 500 °C for 30 min. A MAPbI_3_ perovskite solution (1.0 M) in anhydrous DMF:DMSO 4:1 (v:v) was spin coated onto the compact TiO_2_ in one-step program at 4000 rpm for 30 s. One hundred microliters of chlorobenzene was poured on the spinning substrate 18 s prior to the end of the program. Then, the perovskite film was annealed using a multistep program at different temperatures and times. The hole transport layer of Spiro-MeOTAD was then deposited by spin-coating at 4000 rpm for 30 s. The spin-coating formulation was prepared by dissolving 72.3 mg Spiro-MeOTAD, 28.8 μL 4-tert-butylpyridine (tBP), 17.5 μL of a stocking solution of 520 mg/mL (45 µL 170 g/mol) lithium bis(trifluoromethyulsulphonyl) imide (Li-TFSI) in acetonitrile and 29 μL of a stocking solution of 300 mg/mL tris(2-(1H-pyrazol-1-yl)-4-tert-butylpyridine cobalt(III) bis(trifluoromethylsulphonyl) imide (FK 209 Co(III) TFSI Salt) in acetonitrile in 1 mL cholorobenzene. Finally, 120 nm of silver was thermally evaporated on the top of the device at 1 Å/s when pressure became lower than10^-6^ mbar.

### 3.3. Characterisation

The morphology of perovskite films and cross-sections of different layers on FTO glass were characterized using scanning electron microscopy (SEM, ZEISS EVO/LS15, Oxford, UK) and atomic force microscopy (AFM, NanosurfEasyscan, Germany). FTIR was recorded on an FT-IR spectrometer (Perkin Elmer, L1280026). UV-vis absorption and transmittance spectra were measured using a UV/Vis/NIR spectrometer (Perkin Elmer, Lambda 750S, Waltham, MA, USA). The thickness of compact TiO_2_ layer was measured using a profilometer (Bruker, Coventry, UK). X-ray diffraction (XRD) pattern was characterized with a X-ray diffractometer (D8 Discover, Bruker, UK) at a working voltage of 40 kV and working current of 20 mA using Cu K_α_ X-radiation (λ = 1.5418 Å). The photoluminescence (PL) spectrum of perovskite film was recorded on spectrometer (Lifespec-PS, Edinburgh Instruments, Livingston, UK) using a picosecond pulsed diode laser with a wavelength of 405 ± 10 nm.

Solar characterizations: Photocurrent density–voltage (J–V) characteristics were measured by a computer-controlled digital source meter (Keithley 2400) under ~1 sun illumination. Irradiation of 88.35 mW cm^−2^ light was made with a Sol1A class ABB solar simulator (Model 94021A, Oriel Instruments, Irvine, CA, USA). A thin black mask which was used to define the active area of all photovoltaic devices tested to be 0.05 cm^2^. Bias voltage scanning was carried out at step voltage of 7 mV from −0.2 to 1.2 V at a scanning rate of 35 mV s^−1^. Electrical impedance spectroscopy (EIS) measurements were performed using a potentiostat (Interface1000E, Gamry Instruments, Warminster, UK) on the perovskite solar cells under dark conditions. A dc potential bias was applied and overlaid by a sinusoidal ac potential perturbation of 10 mV over a frequency range from 1.5 MHz to1 kHz. The applied dc potential bias was scanned over 500 to 0 mV in ∼50 mV steps.

## 4. Conclusions

In this work, the role of ethanol in the recipe of TiCl_4_ solution has been quantitatively probed by varying the treating time and analyzing the surface morphology (surface roughness and pore size), and the thickness of ETL TiO_2_ on the photovoltaic performance has also been quantitatively studied. It was found that the ethanol in the recipe of TiCl_4_ solution could efficiently inhibit the agglomeration of TiO_2_ nanoparticles and control the surface roughness and pore size of TiO_2_ film. The origin of the best device performance based on the intermediate thickness of circa 152 nm arises from the relatively smallest charge transfer resistance using Nyquist plot. The champion efficiency beyond 19% of planar perovskite solar cells was achieved via TiCl_4_ water/ethanol solution treatment of TiO_2_ electron transport layer for 30 min at 70 °C. This work underscores the promising potential of the simple planar PSCs. We believe that more efficient and stable planar-structure perovskite can be obtained by further optimization of each layer and interface in the device, such as surface/bulk passivation, compositional, and interface engineering.

## Figures and Tables

**Figure 1 molecules-24-03466-f001:**
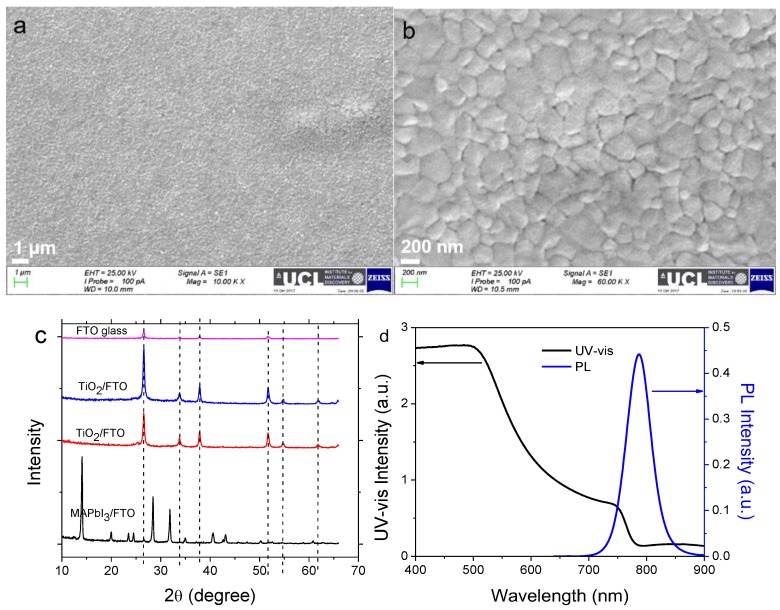
(**a**,**b**) scanning electron microscopy (SEM) images of MAPbI_3_ annealed on fluorine-doped tin oxide (FTO) glass, (**c**) X-ray diffraction (XRD) pattern of perovskite MAPbI_3_ and TiO_2_ on FTO glass and (**d**) UV-vis and photoluminescence (PL) spectra of MAPbI_3_ perovskite on a bare glass substrate.

**Figure 2 molecules-24-03466-f002:**
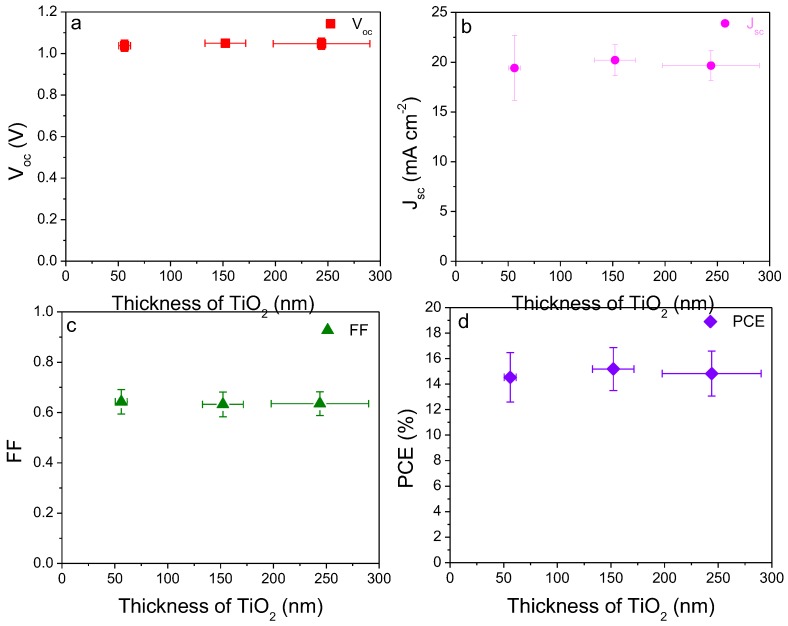
Photovoltaic parameters as a function of thickness of the electron transport layer (ETL) TiO_2_. (**a**) open-circuit voltage (*V*_oc_) (**b**) short-circuit current density (*J*_sc_), (**c**) fill factor (FF) and (**d**) power conversion efficiency (PCE).

**Figure 3 molecules-24-03466-f003:**
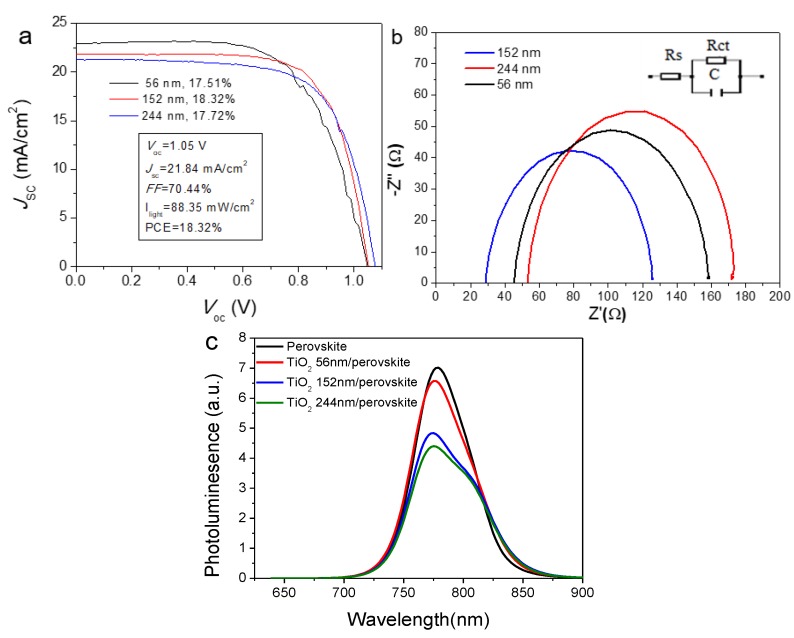
(**a**) J-V curves, (**b**) electrical impedance spectroscopy (EIS) curves of the assembled perovskite solar cells, and (**c**) PL spectra of MAPbI_3_ perovskite on bare and TiO_2_ deposited glass substrate with various thicknesses of 56, 152, and 244 nm.

**Figure 4 molecules-24-03466-f004:**
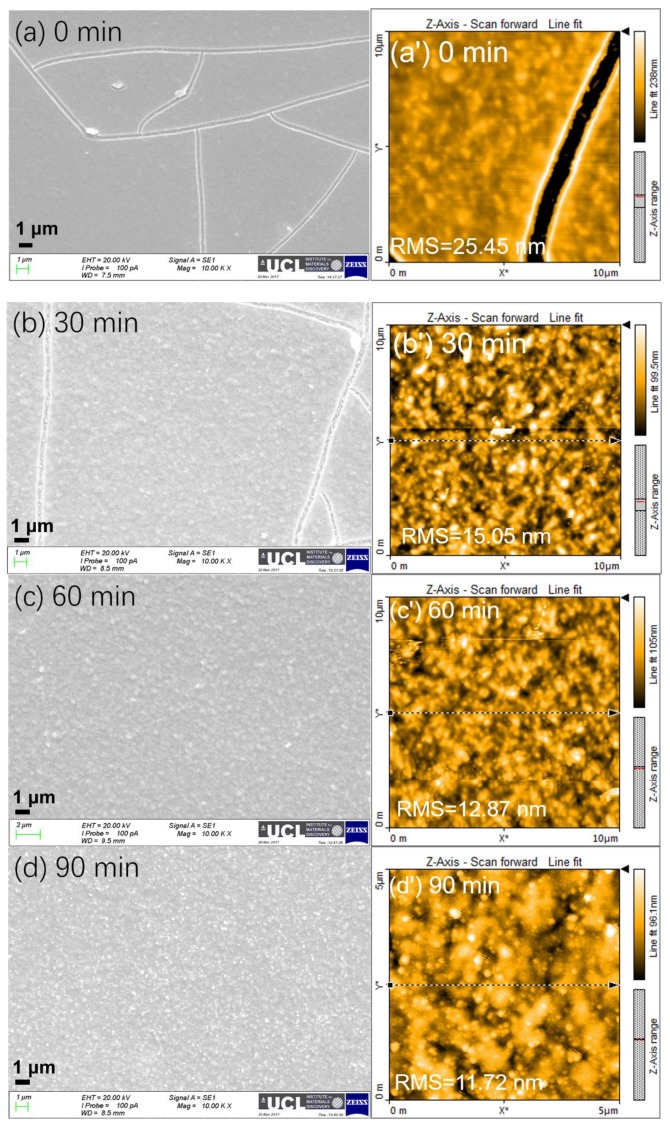
SEM and AFM images of blocking layer TiO2 deposited on after treatment time of TiCl4 for different times varying from 0, 30, 60, to 90 min. a, b, c and d represent SEM images while a’, b’, c’ and d’ denote AFM images of samples pretreated for 0, 30, 60, to 90 min, respectively.

**Figure 5 molecules-24-03466-f005:**
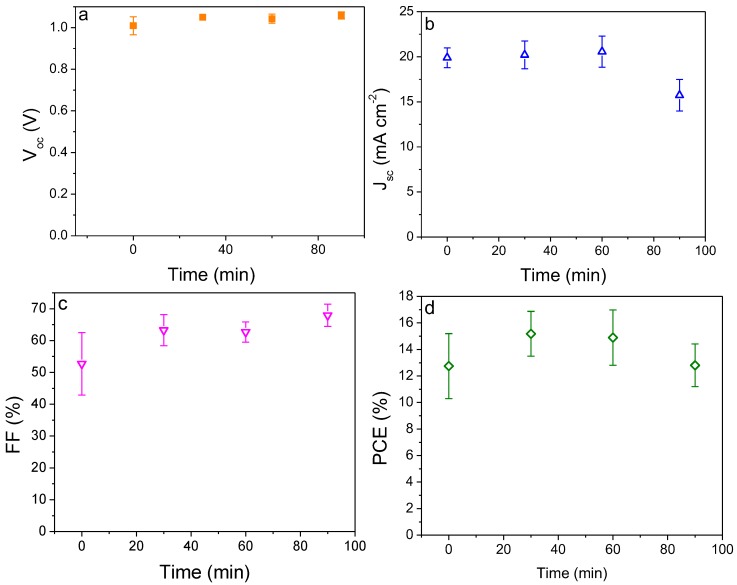
Photovoltaic parameters as a function of electron transport layer TiO_2_ treated by 40 mM titanium tetrachloride for different times: (**a**) *V*_oc_, (**b**) *J*_sc_, (**c**) FF and (**d**) PCE.

**Figure 6 molecules-24-03466-f006:**
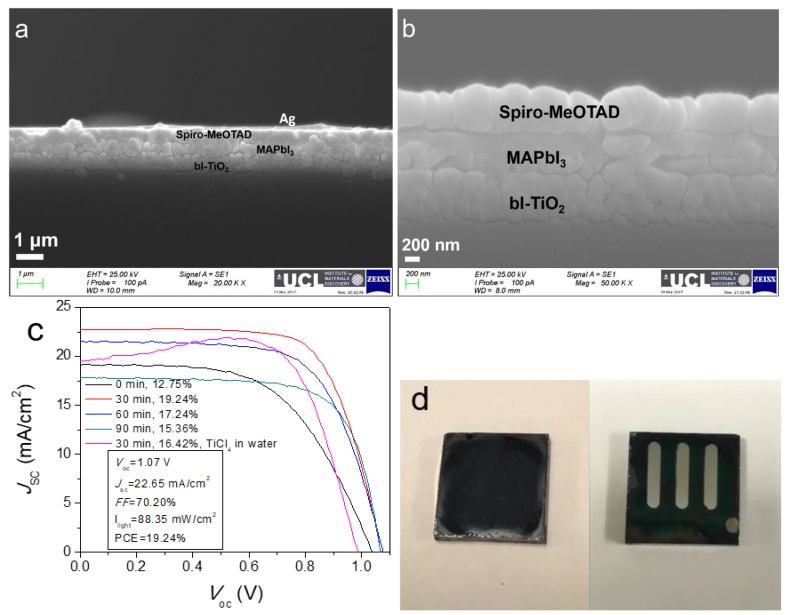
Cross-sectional SEM images at low and high magnifications (**a**,**b**) and typical J-V curves (**c**) of perovskite solar cells with a typical configuration of FTO/TiO_2_/MAPbI_3_/Spiro-MeOTAD/Ag and (**d**) digital pictures of thermally annealed perovskite film on TiO_2_/FTO glass and one PSC device.

**Figure 7 molecules-24-03466-f007:**
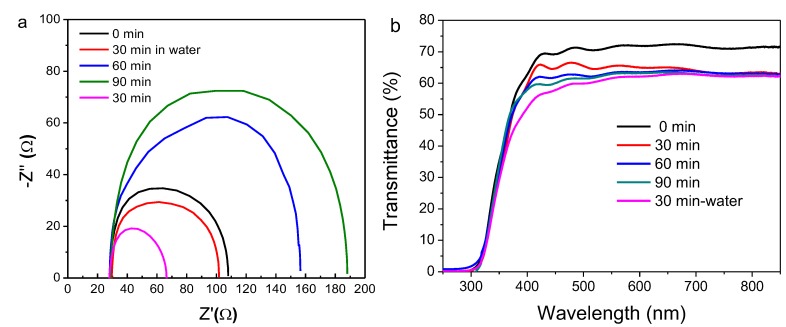
EIS curves of the assembled PSCs (**a**) and transmittance of TiCl_4_ treated TiO_2_/FTO glass (**b**). The applied dc potential bias was changed by ∼50 mV steps from 500 to 0 mV.

**Table 1 molecules-24-03466-t001:** Photovoltaic parameters of the assembled planar PSCs with a typical configuration of FTO/TiO_2_/MAPbI_3_/Spiro-MeOTAD/Ag.

	Thickness ^a^ (nm)	*V*_oc_(V)	*J*_sc_ (mA cm^−2^)	FF (%)	PCE (%)
#1	56.10 ± 5.60	1.04 ± 0.03	19.42 ± 3.25	64.25 ± 4.86	14.53 ± 1.94
Champion		1.05	22.95	64.33	17.51
#2	152.20 ± 19.40	1.05 ± 0.01	20.21 ± 1.54	63.24 ± 4.90	15.18 ± 1.69
Champion		1.05	21.84	70.44	18.32
#3	244.0 ± 46.10	1.05 ± 0.03	19.66 ± 1.48	63.54 ± 4.75	14.82 ± 1.76
Champion		1.07	21.26	68.56	17.72

^a^ Values denoted as mean ± σ, σ is standard deviation, measured from profilometer.

**Table 2 molecules-24-03466-t002:** Photovoltaic parameters of the assembled planar PSCs with a typical configuration of FTO/TiO_2_/MAPbI_3_/Spiro-MeOTAD/Ag.

	Time (min)	*V*_oc_ (V)	*J*_sc_ (mA cm^−2^)	FF (%)	PCE (%)
#1	0	1.03 ± 0.01 ^a^	19.39 ± 0.70	51.74 ± 3.71	11.70 ± 1.00
Champion		1.04	19.14	56.70	12.75
#2	30	1.06 ± 0.01	20.19 ± 1.56	63.67 ± 3.16	15.28 ± 1.60
Champion		1.07	22.65	70.20	19.24
#3	60	1.06 ± 0.02	19.82 ± 2.28	63.00 ± 5.00	14.89 ± 2.08
Champion		1.08	21.54	65.71	17.24
#4	90	1.06 ± 0.02	15.73 ± 1.74	67.92 ± 3.48	12.80 ± 1.61
Champion		1.07	17.80	71.40	15.36
#5^b^	30	0.98 ± 0.01	16.26 ± 2.54	69.64 ± 5.18	12.55 ± 1.89
Champion		0.99	19.48	75.42	16.42

^a^ values are denoted as mean ± σ, σ is standard deviation; ^b^ ETL TiO_2_ layer was treated by TiCl_4_ aqueous solution.

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
