# Peer review of "The Role of Thickness Control and Interface Modification in Assembling Efficient Planar Perovskite Solar Cells"

_molecules, 2019, doi:10.3390/molecules24193466_

Round 1

Reviewer 1 Report

Comments to the Author

The manuscript proposes a facile and attractive method of TiCl4 post-treatment on as-prepared compact TiO2 layer, obtaining dramatic increase of power conversion efficiency for the conventional MAPbI3-based perovskite solar cells in planar structure. Several dominant factors have been effectively tuned to optimize the device performance such as thickness of c-TiO2 layer and the post-treatment time of TiCl4. Eventually, a high PCE of 19.24% has been achieved after optimizing those key factors, indicating that the quality of this work is overall quite high. However, I still feel several concerns about publishing this manuscript from the interpretation of presented data in this work, and recommend publishing in Molecules if authors can consider below comments.

 A couple of major comments:

1.      Thickness of c-TiO2 layer:

a.       Authors have tuned the thickness of c-TiO2 layer ranging from thin ~50 nm to thick ~250 nm, proving that charge transport resistance Rct shows smallest value when medium thickness of 152 nm based on the EIS results. However, as my understanding the charge transport resistance is normally varying proportionally with the variation of ETL thickness. Also, from J-V curves of different thickness based devices (Fig. 3a), it is clearly seen that thinnest c-TiO2 device obtains highest Jsc due to more efficient charge separation and transport. Authors should discuss more or acquire additional data to address this concern, i.e. photoluminescence quenching experiment for c-TiO2 thickness dependent films (glass/c-TiO2/perovskite) compared to pure perovskite (glass/perovskite).

b.      Based on the SEM image shown in Fig. 6b, the thickness of c-TiO2 layer is more than 400 nm, but authors have only claimed the range up to 300 nm. Please clarify.

2.      Morphology after TiCl4 treatment: in Fig 4a and 4b, there are cracks on the surface of c-TiO2 layer, what reasons cause the generation of those cracks? Have authors always seen this sort of cracks on different thickness of c-TiO2 layers? The main purpose of TiCl4 treatment is to passivate the surface state while potentially involving doping effect on TiO2 nanocrystalline film, and to promote the charge separation and charge transfer at the interface of perovskite/c-TiO2. However, authors obtain best performance with 30 min TiCl4 treated device which still shows cracks and relatively higher surface roughness. Please discuss more or add some additional data to address this concern.

3.      Structure dependent charge separation dynamics: it has been reported (e.g. ACS Applied Energy Materials, 2018, 1 (8), 3722-3732 and Journal of Photopolymer Science and Technology, 2018, 31 (5), 633-642) that the electron injection yield is quite low (< 20%) for planar structure cells (c-TiO2/perovskite) compared to relatively high electron injection yield (>90%) for mesoporous structure cells (c-TiO2/m-TiO2/perovskite) since highly efficient hole injection (close to 100% injection yield) dominates the overall charge separation efficiency. Authors have mentioned there is a possibility that some porous TiO2 could form on top of the compact TiO2 layer, and this could indeed influence the interfacial electron injection dynamics. To address this concern, additional data (e.g. PL quenching data) and discussion is highly recommended, to probe the influence of TiCl4 treatment on the interfacial electron injection.

Some other minor comments:

1.      Page 1 Line 27, please provide references when talking about applications of laser diodes and photodetectors.

2.      Page 1 Line 28, “23.3%” has been increased to “24.2%” which was officially reported by NREL, please check and update.

3.      Page 2 Line 45-46, please refer “Chemical Communications, 2016, 52 (4), 673-676” and “Journal of Photopolymer Science and Technology, 2018, 31 (5), 633-642” when discussing about trap state and surface morphology influenced charge recombination kinetics.

4.      Page 6 Line 185-186, please refer “ACS Applied Energy Materials, 2018, 1 (8), 3722-3732” when discussing about the diffusion coefficient and the photoluminescence decay rate.

5.      Several typos should be rectified, i.e. Page 2 Line 61 “PbI2” should be “PbI2”, Page 3 Line 101 “lowerthan” there should be space in between, Page 6 Line 202 “underling” should be “underlying” and etc.

Reviewer 2 Report

In this manuscript the authors report the facile thickness control and post-treatment of the TiO2 electron transport layer to fabricate efficient planar pervoskite solar cells. The TiCl4 is employed to further modify the surface of the TiO2 electron transport layer. In addition, ethanol is also introduced into the TiCl4 aqueous solution when processing. After optimizing the processing conditions, a power conversion efficiency of 19.24% is achieved for pervoskite solar cell. Although the efficiency of the devices is efficient, this work is not suggested to be published in the journal of Molecules before the authors answer the following questions.

1)       Since TiCl4 is commonly used to modify the TiO2 surface in pervoskite solar cells, the motivation of this work is not very clear. Given the authors present ‘’though TiCl4 treatment and thickness of 100-150 nm…the quantitative analysis of their role or the origin has been largely ignored’’ as one of the motivations in this work, the authors should present the answers in the conclusion section.

2)       In the introduction (line 55-57), the authors present the addition of ethanol can control the hydrolysis better. Any data to support this viewpoint? What about the hysteresis J-V curves?  

3)       In the Figure 7b, the transmittance of TiCl4 treated TiO2/FTO glass change with TiCl4 processing time, what about the EQE of the devices at different TiCl4 processing time?

Round 2

Reviewer 1 Report

Comments to the Author

The Authors have provided very positive response and sufficient revisions based on previous comments. Now, I recommend publishing this manuscript in Molecules after the Authors consider below some minor comments.

Page 8 in last paragraph, when discussing about quenching data, the expressions of “becomes much faster” and “PL quenching rate” are not proper as PL quenching is still steady-state data rather than dynamic data. I commend to replace with “PL quenching efficiency” or “enhanced quenching effect” etc.

Author Response

Thank you very much for the review's comment.This is a very good point. We have updated the main text and replaced "becomes much slower "and "quenching rate" with "enhanced quenching effect "and "quenching effect" in the main text ,respectively.